

# Significant water vapor fluxes from the Greenland Ice Sheet detected through water vapor isotopic (δ18O, δD, deuterium excess) measurements

Ben G. Kopec[1], Pete D. Akers[2], Eric S. Klein[3], Jeffery M. Welker[1,4]

[1]Department of Biological Sciences, University of Alaska Anchorage, Anchorage, 99508, AK, USA
[2]Institut des Géosciences et l'Environnement, CNRS, Saint Martin d'Hères, 38400, France
[3]Department of Geological Sciences, University of Alaska Anchorage, Anchorage, 99508, AK, USA
[4]Ecology and Genetics Research Unit, University of Oulu, Oulu, 90014, Finland, and University of the Arctic-UArctic

*Correspondence to*: Ben G. Kopec (bgkopec@gmail.edu)



**Abstract.** The summer of 2019 was marked by an extensive early onset of surface melt and record volume losses of the Greenland Ice Sheet (GrIS), which is part of a larger trend of increasing melt over time. Given the growing spatial extent of melt, the flux of water vapor from the ice to the atmosphere is becoming an increasingly important component of the GrIS

mass balance that merits investigation and quantification. We examine the isotopic composition of water vapor from Thule Air Base, NW Greenland, particularly the deuterium excess (*d-excess*), to quantify the magnitude of GrIS vapor fluxes. To do this, we observe only water vapor transported off the ice sheet (i.e., when easterly winds occur) and during the active melt season. We find that the GrIS-derived water vapor *d-excess* values are controlled by two main factors: 1) the *d-excess* of the sublimating vapor, which is determined, in part, by the relative humidity and wind speed above the ice sheet, and 2) the

proportion of sublimation- vs. marine-sourced moisture. Here, the GrIS melt extent serves as a proxy for the sublimation source and the North Atlantic Oscillation provides a measure of the meridional transport of marine moisture. We demonstrate that sublimation contributes ~20% of the water vapor transported from the GrIS during the melt season. Sublimation is thus an important component of GrIS mass balance and the regional hydrologic cycle, and this flux will become more important in the coming years as further warming continues GrIS negative mass balance trends.

**1. Introduction**

Over the past several decades, the Greenland Ice Sheet (GrIS) has experienced increasing rates of mass loss (e.g., Rignot et al., 2008; Zwally et al., 2011; Hill et al., 2018; Mouginot et al., 2019; Noel et al., 2019; Shepherd et al., 2020). This accelerating melt was marked by record mass loss during the summer of 2019 (Duncombe, 2019; Sasgen et al., 2020), with particularly extreme loss in northern Greenland (Velicogna et al., 2020). Investigating the drivers and components of ice sheet

mass balance is critical in order to determine how this ice loss will impact the planet, particularly through its contribution to sea level change. Many mass balance components, such as surface meltwater runoff and direct ice discharge, are now monitored continuously and at high spatial and temporal resolutions (e.g., Enderlin et al., 2014; Smith et al., 2015; van den Broeke et al., 2017; Trusel et al., 2018), and have led to great improvements in our understanding of the changing ice sheet mass balance. However, a lesser studied component of mass balance is that of water vapor fluxes, specifically those of mass

loss from sublimation of surface snow and ice (e.g., Box and Steffen, 2001; Cullen et al., 2014; Boisvert et al., 2017). While typically estimated to be a much smaller flux than other components (e.g., Box et al., 2006; Fettweis et al., 2017; King et al., 2020), sublimation vapor fluxes are still significant to ice sheet mass balance. This importance requires better quantitative estimates and constraints on vapor fluxes, especially in a changing climate system and the amplification of the Arctic water cycle (Vihma et al. 2016).

In addition to the direct effect on mass balance, these water vapor fluxes can have a critical effect on the local hydrologic cycle through the moistening of the Arctic atmosphere. In the summer months, when incoming solar radiation and air temperature are highest, there is mass loss by sublimation even at the highest altitudes of the ice sheet (Cullen et al., 2014). Near the center of the ice sheet at Summit, Greenland, it has been suggested that vapor arriving from sublimation elsewhere



on the ice sheet contributes a significant proportion (~20%) of moisture to summer precipitation (Kopec et al., 2019). In the

ablation zone of the ice sheet, it is likely that sublimation fluxes are much greater and could also contribute relatively large proportions of moisture to a given air mass.

Back-trajectory analyses around Greenland show that a significant proportion of moisture that contributes to air masses and precipitation on the island is sourced from land within Greenland itself (e.g., Steen-Larsen et al., 2013; Bonne et al., 2014, 2015; Nusbaumer et al., 2019). These land-based sources are often overlooked in favor of the more dominant

surrounding marine sources, but it is possible these models are accurately identifying fluxes of moisture from the GrIS itself. As many of the same meteorological conditions that drive melt on the ice sheet also enhance sublimation and evaporation (e.g., incoming solar radiation and warm air temperature), it can be expected that their potential for moisture contribution also increases given the accelerating summer melt of the ice sheet (Boisvert et al., 2017).

A particularly powerful tool to examine water vapor fluxes is through the measurement of the isotopic ($\delta^{18}O$, $\delta D$,

deuterium excess) composition of water vapor (e.g., Steen-Larsen et al., 2013; Bonne et al., 2014; Klein et al., 2015; Klein and Welker 2016; Akers et al., 2020). Recent technological developments to measure water vapor isotope ratios at high frequency have advanced our understanding of water entering the vapor phase, mainly through evaporation from marine and lacustrine sources (e.g., Kurita, 2011; Benetti et al., 2014, 2017; Steen-Larsen et al., 2014a, 2015; Feng et al., 2016; Bonne et al., 2014, 2019). The vapor isotopic composition has been harnessed in these and other studies to trace hydrologic processes and to

partition moisture from various sources, especially through the measurement of deuterium excess (*d-excess*), a second order parameter that reflects the relationship between the oxygen and hydrogen atoms (e.g., Kopec et al., 2014; Bonne et al., 2015, 2019; Steen-Larsen et al., 2015; Kurita et al., 2016; Akers et al. 2020). Most of the investigations using these new vapor measurements have focused on the liquid to vapor transition, but the snow/ice to vapor phase change has been studied much less (e.g., Steen-Larsen et al., 2014b; Bonne et al., 2019; Madsen et al., 2019), and many questions remain unanswered.

Understanding the isotopic properties of vapor derived from ice is important for quantifying sublimation from the ice sheet, as well as addressing a number of other important questions that utilize isotopic measurements, such as how this vapor exchange between the solid and the gas phase of water may impact the isotopic ratios recorded in ice cores (e.g., Madsen et al., 2019). In this study, we will focus on the *d-excess* values of these vapor fluxes from the ice sheet.

While determining the general *d-excess* values of sublimated vapor is still an active area of research, as some studies

have shown inconclusive evidence for whether the *d-excess* is generally high or low (e.g., Steen-Larsen et al., 2014b), we argue here that the *d-excess* of sublimated vapor is relatively high when compared with moisture evaporated from marine sources. First, from early laboratory studies on sublimation of snow, it is observed that the *d-excess* values of the snow decrease with mass loss under non-equilibrium conditions (Moser and Stichler, 1974). This means that the corresponding vapor, particularly that of the first sublimated vapor, must be relatively high compared to that of the snow. From these simple mass

balance experiments, it can be expected that the *d-excess* of the vapor removed from the snow must be higher than the initial snow/ice. Since that snow is likely primarily sourced initially from marine evaporation, the vapor sourced from these masses should have a *d-excess* greater than that of the marine sourced vapor. In the case of the study by Moser and Stichler (1974),



we can calculate the *d-excess* of vapor lost from the snow, where, for the first 10% of mass lost, the sublimating vapor has a *d-excess* of 33.4‰ (Kopec et al., 2019), which is much greater than the original *d-excess* of the snow that was 6.5‰. Recent studies of water vapor isotope ratios have shown consistency with the hypothesis that sublimating vapor has a relatively high *d-excess*. For example, Madsen et al. (2019) show that there is a relationship between *d-excess* and the latent heat flux, such that when there is net sublimation, the *d-excess* is generally higher than when the latent heat flux shows net deposition/condensation. In another study, Bonne et al. (2019) examine potential sublimation of snow on top of sea ice and show that as sea ice coverage increases, so does the *d-excess*. This finding of increasing *d-excess* with greater sea ice extent suggests that as sea ice coverage increases, the contribution of moisture sourced from the ocean is reduced and the sublimation contribution increases. Several other recent studies also suggest that the primary contribution of sublimation is high d-excess vapor due to significant kinetic fractionation (Kopec et al., 2019; Pang et al., 2019; Bonne et al., 2020). Based on these analyses, we anticipate that as the contribution of vapor fluxes from the ice sheet increases relative to that of marine sources, the *d-excess* also increases.

In addition to sublimation-sourced vapor, it is possible that evaporation of meltwater at the surface of the ice sheet also provides a significant moisture source. This vapor may have a different signal than that of sublimated vapor, but we anticipate the *d-excess* of this vapor is also relatively high compared to marine-sourced vapor given its meltwater source that typically falls near the meteoric water line (or *d-excess* = 10‰) and is evaporating under conditions causing high kinetic fractionation (e.g., Gibson et al., 2008; Feng et al., 2016; Kopec et al., 2018; Cluett and Thomas, 2020). These two GrIS derived vapor fluxes work in tandem to increase the *d-excess* of vapor transported from the ice sheet. While this moisture source is likely increasing over time as the coverage of surface meltwater expands (Leeson et al., 2015), supra-glacial lakes near the margins only reach <2% coverage of the ice sheet surface in recent years and for a relatively short duration during the peak of the melt season (e.g., Sundal et al., 2009; Williamson et al., 2017; Gledhill and Williamson, 2018; Yuan et al., 2020). Given this currently quite small moisture source as compared to the entire snow/ice surfaces available for sublimation, we will focus on the sublimation-sourced vapor for most of the discussion here, but inherently incorporated in that vapor is some smaller amount of meltwater evaporated moisture that has a similar signal.

A new water vapor isotope dataset presented by Akers et al. (2020) from Thule Air Base, NW Greenland (76.5142° N, 68.7442° W), offers a great opportunity to study these vapor fluxes from the ice sheet. At this High Arctic location in coastal northwest Greenland, several hydrologic controls on the isotopic composition of water vapor have been identified and their effects constrained, allowing any additional signal from ice sheet sublimation to be more readily detected. Akers et al. (2020) identify five primary factors that control the vapor isotopic composition at Thule – local air temperature, local marine moisture availability, the North Atlantic Oscillation (NAO), the surface wind regime (i.e. katabatic or sea breeze regime), and land-based evaporation/sublimation. For example, the pressure changes associated with the NAO affect moisture transport patterns over Baffin Bay; whereby during the negative phase of the NAO, this transport from the south is greatly enhanced, which brings in relatively low *d-excess* vapor from this local moisture source. Given these known factors and their effects, we can isolate specific conditions to use the isotopic measurements as a tracer of vapor fluxes from the GrIS. For this particular





analysis, we restrict our water vapor observations to when there are katabatic winds and surface melt is occurring in order to identify the sublimation contribution. By limiting analyses to these conditions, we can more effectively examine the isotopic variations resulting from sublimation contributions and how the proportion of this contribution responds to different transport
patterns associated with the NAO.

In this study, we examine measurements of the isotopic composition of water vapor presented by Akers et al. (2020) from Thule, Greenland from August 2017 through August 2019 to monitor water vapor fluxes from the GrIS, with a particular focus on time periods of significant GrIS melt. As discussed above, for the purpose of this analysis, we assume that the *d-excess* of water vapor sourced from the ice sheet is significantly higher than that of marine sourced moisture, such that an
observed increase in water vapor *d-excess* may suggest an increase in the contribution of water vapor from the ice sheet. The main hypothesis we will focus on in this study is that as melt extent on the GrIS increases, the contribution of moisture from sublimation of snow/ice increases, reflected in higher *d-excess* values. We will examine variations in *d-excess* in order to: 1) identify the signal of water vapor fluxes during melt events, and 2) assess the contribution of moisture from sublimation of snow and ice from the Greenland Ice Sheet.

**2. Data and methods**

**2.1 Water vapor isotope measurements**

A Picarro L2130-i with a Standards Delivery Modulate (SDM) has been monitoring the isotopic composition of water vapor at Thule, Greenland, continuously starting in August 2017. The instrument was installed in a temperature-controlled building on South Mountain (SMT) (76.5142° N, 68.7442° W, 229 m a.s.l.; Figure 1) and air was sampled through ~5 m of
Bev-A-Line IV EVA tubing (3/4 inch diameter) from a collection point above the roof of the building. Local transport is unimpeded by the building or topographic barriers in any direction.

From August 2017 through August 2019, the L2130-i continuously measured $^{18}O/^{16}O$ ($\delta^{18}O$) and D/H ($\delta D$) isotopic ratios approximately once per second. Deuterium excess values (or *d-excess*, where *d-excess* = $\delta D$ - $8\delta^{18}O$) were calculated from those measurements. Calibrations were performed with the SDM using two water standards – USGS 45 ($\delta^{18}O$ = -2.24‰,
$\delta D$ = -10.3‰) and USGS 46 ($\delta^{18}O$ = -29.80‰, $\delta D$ = -235.8‰). These calibrations were conducted approximately every 25 hours, and the isotopic ratios are expressed in the $\delta$-notation as the per mil difference from the VSMOW reference in the VSMOW-SLAP scale. The average standard error of these calibrations was ±0.04‰ and ±0.3‰ for $\delta^{18}O$ and $\delta D$, respectively.

Humidity correction curves were developed for this instrument in summer 2019 to correct for accuracy and precision loss at low vapor concentrations (Steen-Larsen et al., 2013; Bastrikov et al., 2014; Bailey et al., 2015) and applied to the entire
isotopic record [see Akers et al. (2020) for a detailed explanation of the calibration protocol and data quality assurance]. As we will discuss in Sect. 2.3, data selected for this analysis are during the summer months when humidity is relatively high (all hourly $H_2O$ concentrations are >5500 ppm), and at this humidity, the confidence interval of the humidity correction range from



±0.09‰ and ±0.6‰ for δ$^{18}$O and δD, respectively. After calibration, isotopic data were aggregated into hourly means (~3600 measurements per hour) for the analysis in this study.

### 2.2 Climatological conditions

#### 2.2.1 Local meteorology

Wind speed and direction were measured at the Thule Airport (THU; USAF, 2019), a station less than 1 km north of SMT but 170 m lower in elevation. As demonstrated by Akers et al. (2020), meteorological variables measured at both SMT and THU are strongly correlated, and we assume that the wind measurements at THU are similarly representative of conditions at SMT. There is the potential for inversion conditions that separate the air masses in the valley and on top of SMT, leading to katabatic conditions at SMT while THU reports a sea breeze. Thus, some observations during actual katabatic conditions may be excluded from the final database, but they are likely so few in number to make little overall impact to our findings. More importantly, while the opposite error of a katabatic observation at THU while SMT actually has a sea breeze could theoretically affect our results, this is highly implausible due to the climatology and local topography.

For observing conditions on the ice sheet itself (Fausto et al., 2016, 2020), meteorological measurements were also collected from the PROMICE automated weather station at THU_L (76.3998° N, 68.2665° W, 570 m a.s.l; Fausto and van As, 2019), a site 17.8 km east-southeast of SMT on the GrIS. Measurements over hourly intervals at this location, including relative humidity, wind speed and direction, and latent heat flux, are also aggregated for comparison with SMT and THU data.

#### 2.2.2 Greenland Ice Sheet melt extent

As described previously, we anticipate that conditions producing significant vapor fluxes from the GrIS correspond with those that lead to significant ice sheet melt. Therefore, we use the ice sheet melt extent as a proxy for the sublimation source area. GrIS melt extent data are provided by Greenland Ice Sheet Today (Mote and Anderson, 1995), which is produced by the National Snow and Ice Data Center (NSIDC) and based on satellite-based brightness data analysis (Mote, 2014).

In this analysis, we use the melt extent for the entire GrIS as a proxy of the sublimation source area despite the fact that the water vapor we measure in Thule is not sourced from all over Greenland. Instead, there are likely limited regions where this vapor will preferentially sublimate and be transported to Thule. Selecting these specific regions of the ice sheet to better delineate the melt extent for that source area could potentially lead to stronger or different relationships between melt extent and the *d-excess* of water vapor. However, this would require either a subjective decision of delineating that source area or substantial additional analysis (e.g., perform hourly back-trajectory and moisture source identification analyses) to identify for each hour where dominant moisture pathways exist and select the melt extent under those pathways.

We thus use the Greenland-wide melt extent as a proxy of, and not a direct representation of, the sublimation source area. The sublimation source area is likely larger than the melt extent, but we expect melt extent to be a good relative proxy because these two areas would likely be strongly correlated since they both respond to many of the same climatological factors.





The melt extent changes in the northwest region of Greenland have largely followed that of the entire ice sheet, in particular
in capturing the accelerating mass loss contributions and growing ablation zone over the past two decades (Noel et al., 2019).
In using the Greenland-wide extent, we also intend that the results shared in this study are more applicable and comparable to
other studies in different regions of the ice sheet and/or by other researchers examining these GrIS vapor fluxes.

Daily melt extent data were linearly interpolated to hourly resolution, with each hourly observation representing melt
extent of the preceding 24 hours. As this 24 hour period spans both the current and previous day in most cases, the hourly melt
extent values are the average of both days' reported extents, weighted by proportion of each day that makes up the 24 hour
period.

### 2.2.3 North Atlantic Oscillation

To examine how variations in moisture transport affect the isotopic composition at Thule, we examine the NAO as a
proxy for transport conditions. For this analysis, we use the NAO index computed daily by the National Oceanic and
Atmospheric Administration (NOAA) Climate Prediction Center. Akers et al. (2020) show that the maximum influence of the
NAO is significantly ($p < 0.05$) expressed two days later on a *d-excess* measurement and, although not significant at a $p < 0.05$
level, still explains more variance than any other lead/lag combinations at zero and one day lags. For this reason, we compute
an average NAO value that is representative of the 48 hour period before a given *d-excess* measurement in the same manner
as we compute the scaled-up melt extent.

As these transport changes associated with the NAO are driven by large-scale atmospheric pressure changes, we also
examine the spatial variations of pressure around Greenland. To do this, we use NCEP/NCAR Reanalysis (Kalnay et al., 1996)
sea level pressure (mb) data and observe the pressure anomalies over and around Greenland.

### 2.3 Data selection

To most effectively identify and examine the water vapor fluxes from the GrIS, we limited this dataset to only time
windows when vapor transport from the ice sheet to our SMT observation site was favored. Thus, we restrict our analysis to
periods: 1) when the wind direction indicates that transport is coming from the ice sheet, and 2) when significant GrIS surface
melt is occurring so that sublimation fluxes are at their highest potential levels.

The SMT site has a bimodal distribution of wind direction consisting of opposing katabatic and sea breeze regimes
(Akers et al., 2020). For this analysis, we are most interested in the katabatic regime to examine times of transport off the ice
sheet. To identify measurement times under this transport regime, we exclude all data except for when the wind direction is
from the east between 45 to 135 degrees (i.e., from the direction of the local ice sheet). This range of wind directions includes
the mean katabatic wind azimuth at 100 degrees. While it is possible for transport to occur from the ice sheet beyond that
relatively narrow window, this data filter provides the best chance to capture this signal while limiting other moisture sources.
Any hourly observation when the wind direction is from the ice is used in this analysis, even if the surrounding time windows
show wind directions from beyond this narrow scope. As Kopec et al. (2014) showed in Kangerlussuaq to the south of Thule,



there exists a relatively sharp sea breeze front where katabatic winds from the east collide with the sea breeze from the west, separating the marine and glacial air masses. Assuming similar dynamics around Thule, it can be assumed that even short-term shifts to katabatic direction provide representative samples of the air mass derived from the ice sheet.

We also focus on periods when sublimation is at its highest rates during the summer melt season. For this analysis, we define the melt season each year in a similar manner as the Danish Meteorological Institute (DMI), where the melt season starts on the first of three consecutive days where melt occurs on more than 5% of the ice sheet surface, or 85500 km$^2$, and lasts until the final day melt extent is reduced below that same threshold. One difference we make from the DMI definition of the melt season is that we only consider the continuous melt season during the summer, so as to limit the introduction of additional factors that control *d-excess* of water vapor that might occur at the time of spring melt and/or late summer/early fall

melt. The melt season for each of the years in this study extends from early June through mid-August (Table 1).

To restrict our data to times when vapor fluxes represent net mass loss from the ice sheet (i.e. sublimation or evaporation) rather than mass gain into the ice sheet (i.e. deposition or condensation), we only use measurements when latent heat flux at the THU_L site is less than 0 W/m$^2$.

To summarize the criteria that filter the data for the analysis presented in this study, we examine only data that meet

the following conditions:

1)  Wind direction at THU is between 45-135 degrees,

2)  During the continuous melt season when GrIS melt extent is greater than 5% (85500 km$^2$) for more than three consecutive days, and

3)  Latent heat flux at THU_L less than 0 W/m$^2$.

The water vapor isotope measurements that meet the above criteria are hereby defined as GrIS-vapor.

## 2.4 Analytical framework

In the following sections, we will explore the relationships between *d-excess* and GrIS melt extent to identify an atmospheric signal from the water vapor fluxes associated with this melt. As we have described, we anticipate that as the melt extent increases, the sublimation source area proportionally increases, so melt extent is related to the water vapor fluxes from

the ice sheet. These variations in fluxes can be traced using water vapor *d-excess*, where we hypothesize that the *d-excess* increases as the GrIS vapor fluxes increase.

In addition to observing this relationship across the entire melt season, we have particular interest to examine major melt events, such as the large event during July 2019 (Sasgen et al., 2020), to identify if vapor fluxes from the ice sheet are amplified compared to the rest of the melt season. For the purpose of this analysis, we define a "major melt event", or MME,

as any time when the Greenland-wide melt extent exceeds 400000 km$^2$. We classify the onset of these events to be the day when the melt extent begins increasing continuously as it approaches the 400000 km$^2$ threshold, and the end of these events as the time when the melt extent returns below 400000 km$^2$. For the upcoming analyses, we separate these MMEs from the rest of the melt season (dates identified in Table 1).



In the following discussion, we will first examine the hypothesis that the GrIS melt extent is related water vapor
fluxes, which we will identify through the *d-excess* measurements. We will then consider a number of additional factors that
may influence this relationship, such as the NAO, to further constrain this relationship. Once relationships are established, we
will perform a series of sensitivity experiments to estimate the magnitude of these water vapor fluxes and explore their
implications. For any analyses presented here, if p-values are not explicitly stated, statistical significances refer to tests that
yield p-values less than 0.05.

## 3. Results

### 3.1 Characteristics of selected data

Time series data for hourly measurements of δD and *d-excess* from August 2017 through August 2019 are shown in
Figure 2. Isotopic data that meet all selection requirements laid out in Sect. 2.3 are displayed in black. We find that 543 hourly
observations out of 16996 total hourly isotopic measurements meet the data selection requirements.

The daily GrIS melt extent data accompanies the isotopic data in Figure 2. The melt seasons are highlighted in light
red, which start in early-June and end in mid-August over these years (Table 1). There were three MMEs in the 2018 melt
season and four events in the 2019 melt season (Table 1). In the 2017 melt season, all MMEs occurred prior to our sampling
period.

As first described by Akers et al. (2020), both δD and *d-excess* of the full dataset follow the anticipated annual cycle
where in the summer months δD reaches its maximum and *d-excess* its minimum. Imposed on this clear annual cycle are
synoptic scale changes in δD and *d-excess* that can vary over a similar amplitude as the annual cycle itself. These higher-
frequency variations are the focus of this analysis. During the summer melt season, the *d-excess* of GrIS-vapor has a mean ±
standard deviation of 8.8 ± 4.4 which is significantly greater than the *d-excess* (7.5 ± 4.6) of the data that do not meet the
selection criteria for the rest of the melt season.

Daily NAO Index data are also shown in Figure 2. As expected, no clear annual cycle exists for the NAO and high
frequency changes (e.g., weekly) dominate the variance. The mean daily NAO of selected data is -0.16 and ranges from -1.68
to +1.65, encompassing much of its natural range. The variance of the NAO was not evenly distributed across all melt seasons,
as the 2017 and 2019 melt seasons take place under predominantly negative NAO conditions and the 2018 melt season has
relatively positive NAO conditions.

In the following sections, we examine the hypothesized controls of *d-excess* to attempt to identify the signal of GrIS
water vapor fluxes.





## 3.2 Melt extent and deuterium excess

First, we examine the simple relationship between melt extent and *d-excess*. In Figure 3a, we show a linear regression of the 24-hour melt extent against hourly *d-excess*. While the regression is statistically significant (p = 0.048), very little variance in the *d-excess* is explained ($r^2 = 0.007$) by melt extent.

However, if we isolate observations during the MMEs from the rest of the melt season, we see a very clear separation in the data and much stronger relationships (Figure 3b). During both the MMEs (red) and the rest of the melt season, we observe significant relationships between *d-excess* and melt (MMEs: $p < 0.0001$, $r^2 = 0.08$; melt season, MMEs excluded: $p < 0.0001$, $r^2 = 0.26$). While both regression lines are significant and have positive slopes, the slopes of those lines are quite different; the slope during MMEs ($0.0049 \pm 0.001$; slope $\pm$ standard error) is significantly less than the slope during the melt season ($0.027 \pm 0.002$).

In both scenarios, the positive slope of this relationship is consistent with our hypothesis, where we stated that larger MMEs are correlated with greater vapor source areas and fluxes of vapor, and thus higher *d-excess* values. Therefore, it appears that we do indeed observe a significant signal of vapor sourced by sublimation from the ice sheet. It is notable that during the MMEs, the regression slope is much lower, suggesting that the influence of this moisture source is reduced, either in the absolute magnitude of the flux and/or in the proportion of total vapor we are measuring. We explore these options in the following section.

## 4. Controls of water vapor fluxes from the GrIS

While the GrIS melt extent is related to the Thule water vapor *d-excess* values, suggesting that there are significant water vapor fluxes from the ice sheet to the coastal zone, it is clear that additional factors control the magnitude of these fluxes. The separation in the relationship between melt extent and *d-excess* during MMEs and the rest of the melt season can yield insights into these controlling factors.

Here, we propose that the observed *d-excess* at Thule is controlled by two main factors: 1) the *d-excess* of sublimated water vapor, and 2) the proportion of moisture sourced by sublimation as compared to marine evaporation (i.e., mixing sources). We argue that the *d-excess* of sublimating vapor is largely determined by the amount of kinetic fractionation as vapor leaves the ice sheet. This latter factor, the proportion, is driven by two key components: 1) the amount of sublimated vapor available, where melt extent serves as a proxy for the sublimation source area and its vapor contribution, and 2) the transport of external marine-sourced moisture, where we examine the NAO as a proxy for this transport. We explore these key factors in Figure 4 to illustrate how they combine to determine the *d-excess* that is measured in Thule.

In what follows, we further examine these factors from a theoretical perspective and then test these hypotheses empirically.

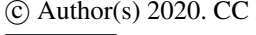



## 4.1 Deuterium excess of sublimation-sourced vapor

As described previously, we assume that the *d-excess* of sublimated vapor is relatively high compared to marine sourced vapor. While we consider this assumption broadly consistent under most conditions, *d-excess* values can vary substantially due to the degree of kinetic fractionation taking place during sublimation as the vapor is transported away from the surface of the ice sheet. Here, we focus on two key factors that are consistently attributed to vapor *d-excess* control (Klein and Welker, 2016; Akers et al., 2020; Bonne et al., 2019): relative humidity and wind speed.

The relative humidity (RH) of the atmosphere that vapor is diffusing into has long been considered a significant control of the amount of kinetic fractionation, whether during evaporation from the ocean (Benetti et al., 2014) and lakes (Feng et al., 2016) or sublimation from snow and ice surfaces (Bonne et al., 2019). Merlivat and Jouzel (1979) modeled that the *d-excess* of vapor increases as the RH decreases, and this pattern has been seen in observations of the isotopic composition of water vapor across a variety of ocean surfaces (e.g., Uemura et al., 2008; Kurita, 2011; Benetti et al., 2014; 2017). With other factors held constant, vapor diffusing from a water surface across a boundary layer with a strong humidity gradient from the surface to the free atmosphere will experience greater kinetic fractionation, and thus a relatively high *d-excess* compared to a situation with a weaker humidity gradient with otherwise similar conditions. Because the humidity gradient above the surface generally increases as the RH decreases, along with other generally correlating variables with the same effect [e.g., isotopic composition of atmosphere vapor is diffusing into, as demonstrated by Feng et al. (2019)], *d-excess* consistently increases with decreasing humidity. While most modeling and observational efforts examine evaporation over the ocean, the same physical parameters are controlling the *d-excess* of sublimating vapor. We consider here that *d-excess* follows a similar pattern with RH, where the sublimation-sourced vapor has a *d-excess* that increases with decreasing RH.

Another factor we consider as a significant control of water vapor *d-excess* values of this sublimating vapor is wind speed. In modeling the *d-excess* of evaporating moisture, wind speed has been shown to have a number of competing effects (Merlivat and Jouzel, 1979). This complicated role has also been seen in observations of water vapor (Bonne et al., 2019). In our situation, where we are attempting to estimate the *d-excess* of sublimating vapor, we anticipate wind speed to have a dominant effect where higher wind speeds cause higher *d-excess* values whether that is a direct or indirect influence. Here, wind speed represents, in part, the residence time of air over a given space – as wind speeds increase, this time is reduced. More importantly for the *d-excess* of sublimating vapor that enters the free atmosphere, for a given amount of time, as wind speed increases, the amount of external air that flows over a given space increases, air that in the case of katabatic origin have low mixing ratios and very depleted isotopic ratios (Kopec et al., 2014; Bréant et al., 2019). The dominant effect of wind speed is thus that kinetic fractionation increases with wind speed, and thus *d-excess* of sublimated moisture increases with wind speed.

To summarize, we argue that *d-excess* of sublimating vapor increases with decreasing relative humidity and increasing wind speed. In Sect. 4.3, we will explore the empirical relationships between *d-excess* and meteorological observations from the THU_L site on the ice to the southeast of Thule Air Base.





### 4.2 Proportion of sublimation- vs. marine-sourced vapor

In addition to the *d-excess* value of sublimating moisture, another key factor controlling the *d-excess* of katabatic-derived vapor in Thule is the relative proportion of vapor sourced from sublimation relative to marine sources. Here, we consider two main components that determine this proportion: 1) the size of the sublimation source area, and 2) the amount of marine moisture transported over the ice.

As we have discussed previously, many of the same conditions that cause melt on the ice sheet will increase rates of sublimation. Because of this relationship, we consider the GrIS melt extent to be a proxy for the sublimation source area. All other factors held constant, an increase in the sublimation source area yields a greater amount of sublimation-sourced vapor and increases the proportion measured in Thule. Thus, as melt extent increases, the *d-excess* of vapor also increases because of the greater incorporation of sublimation-sourced vapor.

In order for water vapor sourced from a marine location to reach our site in Thule under the constraints put in place for data selection (i.e., where surface winds are blowing from the east), the most likely pathway for this moisture is from Baffin Bay to the south over the western margin of the ice sheet and incorporated into the easterly katabatic winds. A key climate factor that plays a major role in transport in this region is the NAO. The NAO is strongly related to atmospheric transport around Greenland and has been shown to control the isotopic composition of vapor and precipitation at a variety of timescales around the island (Barlow et al., 1993; Sodemann et al., 2008; Nusbaumer et al., 2019). As Akers et al. (2020) note, variations in the NAO are particularly important in northwest Greenland for their control of southerly moisture transport up Baffin Bay, where shifts to the negative phase of the NAO are associated with increased transport of water vapor from Baffin Bay to Thule. This transport is associated with a significant increase in the water vapor concentration, an increase of δD and δ18O, and, most notably, a decrease of *d-excess* due to greater moisture supply from the nearby source of Baffin Bay. Alternatively, shifts to the positive phase of the NAO bring moisture transport predominately from the north around Greenland (Sodemann et al., 2008), which substantially limits local marine sourced moisture reaching Thule, bringing vapor with more depleted δD and δ18O, and high *d-excess* values (Akers et al., 2020).

In addition to the general role of the NAO in controlling moisture transport, it can also play a role in driving surface melt on the GrIS, particularly in the MMEs. Most prominently, a key component of times with substantial increases in melt extent is that they are associated with anomalously strong high-pressure systems that reside over Greenland for extended periods of time (Liu et al., 2016). These pressure anomalies are associated with the same pressure changes that are part of the NAO, and it has been observed that MMEs track the phase of the NAO where the negative phase is associated with high pressure over Greenland and increased melt (Bjork et al., 2018; Bevis et al., 2019). This can be seen in the data covering the years in this study, where the two largest MMEs in the summer of 2019 are associated with shifts to strong negative NAO conditions, while the generally lower melt in 2018 is associated with more positive NAO conditions (although each 2018 MME is associated with a shift towards more negative NAO values).



Because shifts to a negative NAO are associated with MMEs, and the negative NAO causes increased transport from Baffin Bay that brings low *d-excess* moisture, it can be hypothesized that during MMEs, the increased transport of local marine-sourced moisture and an increase in the total water content in the atmosphere would reduce the proportion of vapor from sublimation off the ice sheet recorded at Thule. However, the overall increase of the sublimation source area (as approximated by melt extent) suggests that there is more sublimation-sourced vapor available, and may offset some of the effect of increased marine moisture transport. We explore these competing influences in the following section with our isotopic measurements.

### 4.3 Empirical observations of *d-excess* controls

Here, using our observations in Thule, we explore the hypothesized controls of *d-excess*, including the controls of the sublimation end member and the proportion of marine- vs sublimation-sourced vapor. As described above, we use measurements of RH and wind speed from the THU_L site to the southeast of Thule on the ice sheet to examine the sublimation end member. To investigate the moisture proportion, we consider the 24-hour GrIS melt extent (see Sect. 2.2.2 for definition) to approximate the amount of sublimated vapor available and the 72-hour NAO Index (see Sect. 2.2.3 for definition) as a proxy for the amount of marine-sourced moisture transported to Thule. We present the multiple regression analyses in Figure 5 to examine these four variables as controls of katabatic-derived *d-excess* in Thule.

### 4.3.1 Melt season (excluding MMEs)

First, we examine the *d-excess* relationships to these four factors – RH, wind speed, melt extent, and NAO – during melt season (excluding the MMEs). In Figure 5a, we show the results of a multiple regression of these four factors against *d-excess* that yields a strong overall relationship ($r^2 = 0.45$, $p < 0.0001$) and that all factors are individually strongly significant ($p < 0.001$). Focusing on the variables that drive the sublimation *d-excess* end member, we observe that as relative humidity decreases and wind speed increases, the *d-excess* increases. This is consistent with the hypothesis in 4.1 where lower RH and higher wind speeds are expected to increase the amount of fractionation during sublimation that increases the *d-excess* of the vapor that escapes to the free atmosphere.

In examining the variables controlling the proportion of marine- vs. sublimation-sourced vapor, we observe that higher melt extents and more negative NAO values are correlated with higher *d-excess* values. The positive relationship between melt extent and *d-excess* is consistent with our hypothesis in Sect. 4.2 that higher melt extents are related with a greater sublimation contribution and thus higher *d-excess*s. The NAO, however, is not consistent with the prediction, where we hypothesized that the atmospheric circulation associated with a negative NAO would lead to increased marine-sourced vapor that would lower the *d-excess*. Instead, during the melt season outside of the MMEs, the times of negative NAO are associated with higher *d-excess* values, suggesting a greater contribution of sublimated moisture. We explore this discrepancy more below as we contrast this relationship with that during the MMEs. We also note that, while still strongly significant, the NAO does show a weaker relationship than the other variables, as it only accounts for 6% of the explained variance.





### 4.3.2 Major melt events

Next, we examine the *d-excess* relationships to these four factors during only the MMEs (Figure 5b). Here, we again
see a significant overall multiple regression ($r^2 = 0.39$, p < 0.0001) and significant individual relationships (p < 0.001) between
the *d-excess* and each factor. In Figure 5b, we observe that the relative humidity and wind speed each show similar relationships
with *d-excess* as during the rest of the melt season, where decreasing humidity and increasing wind speed result in increasing
*d-excess* values. Once again, these relationships are consistent with our hypotheses in Sect. 4.1, and the similarities in slopes
to the melt season relationships demonstrate the robustness of this effect. The effect of the GrIS melt extent is also similar in
that higher melt extents are associated with higher *d-excess* values. However, a major difference here is that the slope of this
relationship is significantly lower for the MMEs (0.0046) as compared to the rest of the melt season (0.0193). Another major
difference during MMEs is that the NAO has a significant positive relationship with *d-excess*. This relationship is opposite
compared to the rest of the melt season, but consistent with the hypothesis posited in Sect. 4.2 where transport associated with
negative NAO phases brings marine-sourced moisture that lowers the proportion of sublimation-soured moisture and lowers
the *d-excess*.

Regarding the sublimation *d-excess* end member, it appears that relative humidity and wind speed are predictive
drivers of this value at any time during the melt season and under a range of larger-scale climate conditions. The similar
direction and magnitude of relationships during MMEs and the rest of the melt season provides confidence that these factors
generally have the effects described in Sect. 4.1 on *d-excess* of sublimated vapor through their role in fractionation. That said,
it might also be possible that these relationships could simply be the result of increased sublimation rates, as it would be
expected that relatively drier air and windier conditions may result in higher sublimation amounts. However, the wind speed's
effect on sublimation rates is more complex as higher wind speeds also likely supply more dry glacial air from high on the ice
sheet that could outweigh any contributions from sublimation. In either role, these conditions work together to drive the
observed contribution of sublimation-sourced vapor in Thule.

This analysis also reveals that the GrIS melt extent functions well as a proxy for the contribution of sublimation-
sourced vapor during MMEs. When transport variance is held relatively constant (by including the NAO in the regression) the
d-excess values show that the proportion of sublimation-sourced vapor increases relative to marine-sourced vapor as the melt
extent increases. While an important variable at all times during the melt season, we see the effect of melt extent (i.e., the slope
of the relationship) is much lower during the MMEs. This reduced effect could be the result of a number of factors. In Figure
3, the slope of the relationship between melt extent and *d-excess* is almost 6x less during the MME windows as compared to
the rest of the melt season. This effect is reduced to 4x greater by including the NAO as part of a multiple regression, suggesting
that quantifying the moisture transport effect of the NAO does account for some of this difference. As the NAO does not
effectively quantify all marine-sourced moisture transported to Thule, it is very likely that some residual marine moisture
sourcing effect remains here.





However, additional reasons for why the effect of melt extent is so much lower during MMEs also exist. It is possible that the source area reaches a specific extent where it is so large that any additional effect on the proportion of marine- vs sublimation-sourced vapor is reduced, although the fact that the reduced effect is consistent even from the onset of the MME when melt extent is still relatively low argues against this. Alternatively, other moisture sources, such as meltwater evaporation during MMEs, could provide a significant contribution to the total vapor measured in Thule. Meltwater evaporation would

likely have a relatively high *d-excess* compared to that of local marine-sourced moisture but is lower than sublimated vapor, so it is plausible this source becomes a greater contributor during the MMEs (discussed more in Sect. 5). Additional highly constrained analyses would be needed to tease these two apart.

It is likely that the NAO can play a number of roles in the complex overall water vapor isotope system that change depending on the selected spatial and temporal focus and the antecedent environmental conditions. During MMEs, it appears

that *d-excess* variations are driven mostly by atmospheric transport changes related to the NAO. The strong positive relationship between *d-excess* and the NAO observed in Figure 5b shows that the conditions of the negative NAO phase are driving significant marine-sourced moisture transport from the south to Thule. The relatively low *d-excess* values sourced from Baffin Bay and the North Atlantic (Akers et al. 2020) outweighs the high *d-excess* sublimation-sourced moisture off the GrIS. Although MMEs are typically associated with negative NAO conditions, there are times when these events occur during the

positive phase. On these occasions, we see the expected change in the *d-excess* where this shift to the positive phase results in limited locally-sourced marine moisture, and thus the *d-excess* increases because of the increased proportion of sublimation-sourced vapor as well as the increased *d-excess* of marine-sourced moisture that has arrived from more distant locations and undergone more Rayleigh distillation (e.g., Jouzel and Merlivat, 1984; Ciais and Jouzel, 1994).

Outside of these MMEs, the NAO has the opposite relationship with the Thule vapor *d-excess* during the rest of the

melt season. It is plausible that the negative phase of the NAO could result in increasing the sublimation source area that is not captured by the Greenland-wide melt extent. This would also result in the higher proportion of sublimation-sourced moisture, thus increase the *d-excess*. Whether through its control of the sublimation source area or by other factors associated with the NAO, this effect would likely still contribute to the partial regression of the NAO against *d-excess* during MMEs, but the impact of variable marine-sourced moisture transport depending on the NAO phase overwhelms these effects during MMEs.

On the other hand, the transport effect of the NAO also surely contributes to the partial regression during the melt season, and may be a cause of the weaker relationship as compared to the other variables in the multiple regression in Figure 5a.

### 4.4 Strength of relationships and uncertainties

While the relationships shown in Figure 5 are statistically significant, there are a number of potential sources of uncertainty that contribute both to the unexplained variance in those relationships and uncertainty in the relationship itself.

Certain choices and assumptions can lead to biases or have unexpected effects on the analysis presented here. We consider the relationships in Figure 5 to be robust, but we will explore some of these factors below.



First, we have used *d-excess* as the primary indicator of moisture source changes, as is typically done, but much of this same information can be examined from the $\delta^{18}$O and $\delta$D values of this GrIS-vapor. There are a number of additional complicating factors to consider here, but because of the same mechanisms we have described that lead to the sublimation-

sourced moisture having a high *d-excess* relative to the nearby marine-sourced vapor, and that *d-excess* and $\delta^{18}$O or $\delta$D are typically anticorrelated in these relationships, we would expect that the sublimation-sourced $\delta^{18}$O and $\delta$D values be relatively depleted compared to that of the marine vapor.

Given the influence of these factors on *d-excess* values, we would anticipate that $\delta^{18}$O and $\delta$D values decrease in response to decreasing RH, increasing wind speed, higher melt extent, and more positive NAO. If we perform the same multiple

regressions in Figure 5 with $\delta^{18}$O and $\delta$D values instead of *d-excess*, we see very similar results (Figure S1). During the MMEs ($\delta^{18}$O: $r^2 = 0.42$, $p < 0.0001$; $\delta$D: $r^2 = 0.43$, $p < 0.0001$) and the rest of the melt season ($\delta^{18}$O: $r^2 = 0.28$, $p < 0.0001$; $\delta$D: $r^2 = 0.26$, $p < 0.0001$), all explanatory variables have significant partial regressions in the same expected directions for $\delta^{18}$O and $\delta$D. The only difference is that during the melt season (MMEs excluded), $\delta^{18}$O and $\delta$D decrease as the NAO increases, although this is again the weakest effect of the four explanatory variables only accounting for 5% of the explained variance. This

direction would be consistent with our hypothesis of the effect of marine transport described in Sect. 4.2, but it is the opposite relationship compared to what we see with *d-excess* during time. As we described above, it appears that the NAO is accounting for multiple factors that result in this slightly different (and weaker) relationship. Given the otherwise consistent relationships between the $\delta^{18}$O, $\delta$D, and *d-excess* with the explanatory variables, the hypothesized mechanisms appear to be robust.

In this analysis, we use the melt extent for the entire GrIS as a proxy of the sublimation source area despite the fact

that the water vapor we measure in Thule is not sourced from all over Greenland. Given the consistent strength of the relationships between *d-excess* and melt extent, we conclude this to be a robust assumption. While beyond the scope of this study, more detailed, high-frequency experiments that examine the regional wind field and the sublimation conditions over the northwest sector of the ice sheet would be another valuable contribution.

Sublimation is simply assumed to be a source of high *d-excess* vapor as compared to the nearby marine sources, and,

as presented here, is affected directly or indirectly by the relative humidity and wind speed above the ice sheet, but the isotopic composition of the sublimating vapor can vary because of a number of additional factors. While this vapor will have a relatively high *d-excess* under most circumstances, as we first argue in Sect. 1, the isotopic composition can be altered by the degree of fractionation from the time of the phase-change during sublimation to when we measure the vapor in Thule. These factors could include the initial isotopic composition of the snow/ice, the percentage mass loss of that snow/ice, the isotopic

composition of the vapor over the ice sheet, among others. Sublimation earlier in the melt season will be preferentially from the previous winter's snowfall while later in the melt season it will be more from bare ice. While there might be variability introduced by all of these factors, the strength and robustness of the relationships between RH and wind speed at the THU_L site and the GrIS-vapor *d-excess* measured in Thule during MMEs and the rest of the melt season suggest these variables capture a significant amount of the variability induced by the factors listed above. That said, more measurements on and above

the ice sheet would help to better constrain these relationships. Intensive experiments along the margin of the ice sheet to better

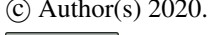


understand the isotopic changes during sublimation, such as in a similar manner to the one performed by Madsen et al. (2019) at the EastGRIP site, would help to reduce these uncertainties.

The rate of sublimation itself also comes with uncertainty in our analysis. We implicitly assume that the amount of vapor supplied by sublimation is proportional to the melt area and/or the fractionation-influenced factors (RH and wind speed).

As with the amount of fractionation, there are clearly other meteorological variables that can affect the magnitude of this vapor flux. There are a number of factors that could also influence the sublimation flux including incoming radiation, air and snow/ice temperatures, snow/ice crystal structure, surface aspect, among others (e.g., Box and Stefen, 2001; Cullen et al., 2014). As with the discussion above regarding the many influences on the degree of fractionation, it appears that the suite of variables used here to predict the GrIS-vapor *d-excess* values captures a significant amount of the variance. However, more extensive

and focused experiments could help reduce uncertainty and allow for a stronger understanding of the processes and conditions that influence sublimation rates.

In this analysis, we focus on sublimation as the dominant flux of water vapor from the GrIS. It is likely that evaporation from meltwater is also a substantial source of moisture when there are large quantities of liquid water at the surface, whether in supra-glacial lakes, surface runoff, or saturated snow/ice, or even in the meltwater rivers and lakes across the ice-

free tundra landscape. Moisture sourced by evaporation will have a different signal than that of sublimated vapor, but we anticipate that this vapor also has a *d-excess* that is relatively high compared to that of water vapor from marine sources. The process of evaporation has a similar effect on the vapor removed from the liquid water as sublimation does with vapor from snow/ice – i.e. the remaining water is enriched and *d-excess* is reduced, while the vapor is depleted with relatively high *d-excess* (e.g., Gibson et al., 2008; Feng et al., 2016). Due to the extent of melt ponds, the effect of these meltwater-based vapor

fluxes is likely inherently incorporated into the relationships we have observed in Figure 5. Given the robustness of the fractionation-related partial regressions, it is likely that these evaporative fluxes have similar relationships as the sublimation-sourced vapor between the meteorological variables and *d-excess*. If the meltwater evaporation moisture has a relatively lower *d-excess* than sublimation-sourced vapor (but still high compared to marine sources), it is possible that the lower slope of the partial regression of *d-excess* against melt extent during the MMEs is caused, in part, by a shift towards more of the GrIS-

vapor sourced from meltwater evaporation. Since the lower slope relationship between melt extent and *d-excess* exists at the onset of these of MMEs, it seems that the change in transport may be the more important factor here as it would seem there would be some lag time between the increasing melt and the evaporation flux, but, nonetheless, the change in GrIS-based vapor flux could have some effect.

It is also possible that water is sourced from other surfaces across the landscape, whether as evaporation from non-

meltwater (meteoric) lakes, transpiration from plants, or evaporation of soil moisture, all of which could have a different isotopic signal. For example, evaporation from meteoric lakes might provide a different vapor signal than from meltwater as it is likely that these lakes have been much more evaporated than the meltwater lakes/rivers and have a lower *d-excess* (e.g., Kopec et al., 2018). However, given the low abundance and coverage of lakes in the periglacial landscape compared to meltwater surfaces on the GrIS and the even larger snow/ice surfaces over the entire GrIS, it would be unlikely for this flux to





be a dominant contributor to the total vapor flux and subsequent *d-excess* values. Given the sparse vegetation coverage across this region (Sullivan et al., 2008), a shallow active layer of up to 1 m, and near-surface bedrock, evapotranspiration is also expected to be quite limited compared to the much more expansive sources from sublimation of snow/ice or evaporation of meltwater.

## 5. Magnitude and implications of GrIS water vapor fluxes

The observations presented here demonstrate that the water vapor fluxes off the Greenland Ice Sheet are an important component of the regional hydrologic cycle. Depending on the magnitude, this source of moisture can have a number of significant impacts on the surrounding landscape, both the ice-free tundra and over the GrIS itself. In the following discussion, we explore several of these implications through examining the role of this moisture source on the hydrologic cycle today and moving forward in a changing climate system, and also consider how this signal may have been recorded in this past.

While a focused investigation at various temporal and spatial scales is needed to truly quantify the vapor fluxes from the ice sheet, we can use hypothetical isotopic end members for the different moisture sources to attempt to examine just how large the contribution of sublimation is to the air masses measured in Thule. Using the same method presented by Kopec et al. (2019), we compute *d-excess* end members for locally-sourced marine vapor for Greenland and sublimation-sourced moisture. Kopec et al. (2019) calculate a sublimation *d-excess* of 33.4‰ from the mass balance experiment presented by Moser and

Stichler (1974). Akers et al. (2020) show that the primary summer marine moisture source is Baffin Bay, so we calculate the mean *d-excess* for vapor sourced here during the summer months to be 2.8‰, based on the source region defined by Kopec et al. (2019) as computed from the isotopically enabled general circulation model ECHAM 4 (Hoffmann et al., 1998). The mean *d-excess* for Thule GrIS-vapor is 8.8‰, so if we use those two end members of marine sourced vapor from Baffin Bay and sublimation as the principle sources, the simple mass balance calculation suggests 20% of vapor is sourced from sublimation.

This contribution is consistent with the estimate by Kopec et al. (2019) for precipitation at Summit, Greenland that is sourced from sublimation. This, again, suggests that sublimation is likely a significant contribution to the water cycle in the summer months. The maximum *d-excess* value measured at Thule for data selected in this investigation is 20.2‰, which, under a situation of a mixture of moisture sourced from Baffin Bay and sublimation, would be equivalent to 57% of the vapor sourced from sublimation. Under the climatological conditions in the time window of this study, this would likely represent the highest

approximate percentage of moisture sourced from sublimation, giving a rough constraint on how large the contribution of this moisture source could be.

    We can do a similar computation to see how this proportion changes in response to variations in GrIS melt. From Figure 5, we observe that, during the melt season (excluding MMEs), for every 100000 km$^2$ increase in melt extent, the *d-excess* increases 1.94‰. This relationship can be converted to a change in the moisture proportion from sublimation versus

marine using the same assumptions above. Thus, a 100000 km$^2$ increase in melt corresponds to an increase in the proportion of sublimation-sourced vapor by 6.3%. During MMEs, this relationship is lower, where for every 100000 km$^2$ increase in melt



extent, the *d-excess* increases 0.46‰. This change in melt extent corresponds to an increase of 1.5% in the proportion of sublimation-sourced vapor.

It is likely some moisture is also sourced from the North Atlantic (Akers et al., 2020). If we, again, follow the guidelines presented in Kopec et al. (2019), the summer North Atlantic subtropical source has an average *d-excess* of 10.6‰. The center of this moisture source at the subtropical high is further south than the primary regions of the North Atlantic-sourced moisture identified in Akers et al. (2020), so this modeled *d-excess* may be higher than the actual *d-excess* of any vapor sourced from the cooler waters of higher latitudes of the North Atlantic. Nevertheless, we can use this end member to estimate the sublimation source contribution, where, if the marine-source contribution is comprised of 50% subtropical and 50% Baffin

Bay-sourced moisture, the sublimation contribution computes to be 8% of the mean Thule water vapor. The 50% North Atlantic contribution to the marine mixture would be a rather high percentage, so this computation can provide a constraint of the low contribution of sublimation-sourced moisture in Thule (under the atmospheric conditions during this study). We can also use this marine moisture end member to compute the sublimation source proportion in relation the GrIS melt extent. During the melt season (excluding MMEs), for every 100000 $km^2$ increase in melt extent, the proportion of sublimation-sourced vapor

increases by 7.2%. During the MMEs, this same change in melt extent corresponds to an increase of 1.7% in the proportion of sublimation-sourced vapor. These rough estimates of the sublimation contribution to ambient vapor suggests that this is a rather significant moisture source in the summer months.

In addition to examining the scale of this moisture source today, we can also consider how the sublimation source area has varied over time. The extent and total amount of melt has increased substantially over the last several decades based

on remote sensing methods (e.g. Sasgen et al., 2020) and on-the-ground measurements such as in ice/firn cores (Graeter et al. 2018). Given the much lower average melt extent and less frequent MMEs ~30 years ago and for decades prior, it is likely that the contribution of sublimation had been relatively limited overall, with the exception of the occasional significant MMEs\. We can again perform rough sensitivity experiments to examine how much the contribution of sublimation has changed based on our estimates in Figure 5. The average GrIS melt extent for the month of July has increased 7750 $km^2$/yr (or ~310000 $km^2$)

from 1979 to 2019 (NSIDC; Mote and Anderson, 1995; Mote, 2014). If we use the same relationship described above for the melt season (MMEs excluded), with a mixture of Baffin Bay and sublimation-sourced moisture, the increase in melt extent over this time period has increased the proportion of the sublimation contribution by almost 20%. This calculated change does not fit the record precisely, as the melt extent for July during our measurement years of 2017-2019 was only ~300000 $km^2$ and so these years fall below the linear trend used above. Nevertheless, if we extend the linear relationship to the beginning of the

observational melt record (i.e. to the 1970s), these rough estimates would suggest that sublimation was likely a relatively insignificant source of moisture at this time. There are likely to be short-term, local conditions that cause significant contributions of sublimated vapor in the summer as the previous winter's snow is ablated at the margins, however, based on this sensitivity experiment, the sublimation source from a larger Greenland-wide perspective was likely quite minimal ~40-50 years ago.



While this sublimation source may not have been a significant source several decades ago, moving forward in a warming Arctic, we expect the contribution of sublimation-sourced moisture to continue to increase. It is expected that as the melt extent continues to increase and the melt season expands beyond the time window we see presently, and with that a greater coverage of supra-glacial lakes and liquid meltwater in general at the surface of the ice sheet (e.g., Gledhill and Williamson, 2018; Noel et al., 2019). Given that we have shown that in all scenarios the sublimation/evaporation contribution increases

with melt, with these projected changes, we expect these water vapor fluxes to continue to increase. Moving forward, these water vapor fluxes may have a greater influence on the regional hydrologic and continue to be a significant net negative mass balance component during the summer months.

Another key observation to understanding the role of the ice sheet as a moisture source is that MMEs appear to have a suppression effect on the proportion of sublimation-sourced moisture in Northwest Greenland due to the nature of moisture

transport associated with the blocking high pressure systems over Greenland. If we separate the MMEs from the rest of the melt season, the average *d-excess* for these two periods is 7.2‰ and 9.6‰, respectively. If we assume the same mixture of Baffin Bay and sublimation moisture sources, the sublimation contribution is only 14% as compared to 22% for the rest of the melt season. It can be expected that this suppression extends southward along the west coast of Greenland as the same marine moisture transport patterns act similarly across this whole region (Nusbaumer et al., 2019). However, while suppression of

water vapor fluxes occurs on the west coast during MMEs, it is plausible the opposite effect occurs on the east coast of Greenland where the dominant transport would be of dry Arctic air from the north, which could result in much higher fluxes of sublimation. Additionally, although vapor fluxes are suppressed during these events, the fluxes still increase with melt extent, just at a lower rate than other periods of the melt season. Therefore, these vapor fluxes have a compounding effect with other features of these MMEs that lead to increasingly negative ice sheet mass balance.

This evolving moisture source has a number of implications. The increased moisture content of this glacially-derived air mass can have a number of impacts on the hydrologic cycle of this region, both on the ice-free landscape and over the GrIS. During the summer months, the wind direction in Thule is most frequently from the west, bring vapor from over Baffin Bay (Akers et al., 2020). This suggests that, unless there is a change of the predominant wind direction, the effect of these water vapor fluxes may be minimal on the ice-free landscape immediately surrounding Thule. That said, during times when the

katabatic winds dominate the region, this typically drying air mass that plays a significant role in the hydrologic and terrestrial ecosystems around Greenland (e.g., Heindel et al., 2015) may be altered as increased moisture is incorporated. While the effect of this moistening air mass might be somewhat limited for the ice-free landscape around Thule, in many other locations around West Greenland where the westerly sea breeze is less dominant and the katabatic winds are the more frequent feature (Kopec et al., 2014), these melt-related water vapor fluxes can have a greater effect on the landscape.

This sublimation-sourced vapor can also have a significant influence on the hydrologic cycle over the GrIS itself. One particular impact could be through its contribution to precipitation on the ice sheet, analogous to terrestrial recycling of water (e.g., Nusbaumer et al., 2019). At Summit, Greenland, near the center of the ice sheet, Kopec et al. (2019) identified the signal of sublimation-sourced moisture in summer precipitation, accounting for ~20-30% of the moisture. As this signal is





seen at high-altitude, interior locations on the ice sheet and near the margins, it is suggestive that sublimation-sourced moisture
does contribute to some degree to precipitation across most of the GrIS in the summer months. The total contribution will
depend on a number of atmospheric conditions at the ice sheet surface, some of which we have explored previously in this
study, and the proportion of moisture is contingent on the transport of this vapor compared to the flux of moisture from marine
sources.

While beyond the scope of this study, the annual cycle of *d-excess* may yield insights to determine how influential
sublimation is as a moisture source in different locations around Greenland. In Thule, we observe the typical Arctic *d-excess*
annual cycle that has relatively low summer *d-excess* compared to high winter *d-excess*. This is a key point because while
sublimation can contribute to high proportions to GrIS vapor in Thule, these air masses are only present for limited times (i.e.
the westerly sea breeze dominates during the summer months; Akers et al., 2020), and so the sublimated vapor has a relatively
low influence on the overall summer vapor in Thule. However, at Summit, Greenland, the *d-excess* annual cycle has flipped
where the summer *d-excess* is relatively high compared to winter, suggesting that this sublimation-sourced vapor is a much
more prevalent feature of the summer hydrologic cycle (Kopec et al., 2019). Presumably, there is a point between the coast
and Summit where there is a flip in the annual cycle as this sublimation moisture source becomes a more significant contributor
to the total vapor. The location(s) where this flip occurs would represent an important threshold where sublimation sourced
moisture becomes a significant contributor to the local hydrologic cycle.

We can similarly use ice cores from around the GrIS to examine melt extent further back in time. Even in the absence
of clear ice layers that represent melt high on the ice sheet, as was observed by Graeter et al. (2018), we can use the water
isotope measurements to examine the sublimation signal and, thus, the extent of GrIS melt. As discussed above in terms of the
spatial variations in the *d-excess* annual cycle and the location where the phase of this cycle flips, we could examine this same
variation temporally through observing the phase changes in ice cores. For example, at Summit, if there is a time when
sublimation stops being a significant contributor to the summer precipitation, the annual cycle would likely flip to have
relatively low summer *d-excess* and become more similar to the coastal locations. The timing of this flip would potentially
suggest the time when the sublimation source area, or the melt extent, is reduced significantly enough to result in an
insignificant amount of sublimation-sourced vapor. This sort of analysis could be conducted at sites all across the ice sheet
(e.g., for a suite of ice cores - Vinther et al., 2010; Graeter et al., 2018), including in sites nearby Thule (e.g., 2Barrel ice core
- Osterberg et al., 2015), to better understand the variations of this moisture source over time.

## 6. Conclusions

In this study, we demonstrate that the flux of water vapor by sublimation from the surface of the Greenland Ice Sheet
(GrIS) contributes a significant proportion of total moisture through water vapor isotope measurements in Thule, NW
Greenland. We restrict the data used in this analysis to focus on only the vapor that has been transported from over the GrIS
prior to reaching our site in Thule (GrIS-vapor). The isotopic signal of this sublimated vapor is distinct from moisture sourced



from marine locations, particularly in its high deuterium excess produced in the highly fractionated sublimated vapor as compared to the relatively low *d-excess* produced during evaporation from nearby marine locations. We observe that the amount of sublimated vapor increases proportionally with the extent of melt on the GrIS, such that the *d-excess* of the water vapor measured in Thule increases significantly with melt extent, both during MMEs and the rest of the melt season.

While *d-excess* is related to the melt extent across the summer months, we show that this relationship varies during MMEs (defined by spikes in melt extent above 400000 km$^2$; see Sect. 2.3) and the rest of the melt season, where the magnitude of this *d-excess*-melt extent relationship is much smaller during the MMEs than the rest of the melt season. This suggests that the incorporation of sublimation-sourced vapor is suppressed during MMEs, either in total amount or at least in proportion compared to marine-sourced moisture.

In addition to the GrIS melt extent, there are a number of factors that are critical to determining the *d-excess* of the GrIS-vapor, all of which are critical to understanding these water vapor fluxes from the ice sheet. The two primary components controlling the *d-excess* measured in Thule are: 1) the *d-excess* of sublimating vapor, and 2) the proportion of sublimation- to marine-sourced vapor. The sublimated moisture *d-excess* is defined, in part, by the amount of kinetic fractionation as the vapor leaves the ice surface and is incorporated into the ambient vapor. Using meteorological measurements at the THU_L site on

the GrIS southeast of Thule, we show that the relative humidity (RH) and wind speed above the ice surface are critical to determining the *d-excess* of the sublimating vapor, such that lower RH and higher wind speeds correspond with relatively higher *d-excess*. The proportion of sublimation- vs. marine-sourced moisture is determined by the amount of sublimated vapor produced and the amount of marine moisture transported to Thule. We show that the GrIS melt extent serves as a proxy for this magnitude of this moisture source, where higher melt extents correspond with higher GrIS-vapor *d-excess*.

The amount of marine transport is shown to be driven, in part, by the changes in circulation associated with the NAO. The NAO, however, is shown to have a complex relationship with this system. During the MMEs, the dominant process affected by the NAO appears to be the change in marine moisture transport, where the negative NAO increases this transport and results in lower *d-excess* values. During the rest of the melt season, this negative NAO is associated with relatively higher *d-excess* values, suggesting it also impacts the sublimating vapor such that negative NAO conditions cause sublimated vapor

of higher *d-excess* and this is the dominant expression during the non-MME parts of the melt season. More detailed studies on both the small scale to examine isotopic changes during sublimation and the large scale to investigate the roles of atmospheric circulation variations will help to better constrain these relationships.

In this analysis, we have shown that sublimation contributes a significant amount of moisture during the melt season and this will have a number of important implications. The moistening of the atmosphere will alter the hydrologic cycle of the

region and affect both the ice-free tundra landscape and the ice sheet itself. As we have discussed, based on similar isotopic observations at higher altitudes on the ice sheet, it is likely that this moisture source is a significant contributor to the hydrologic cycle at most locations around the GrIS during the summer months. Given the current trends and projections in Arctic-wide temperatures and melt extent, we anticipate that the GrIS will continue to be a significant moisture source and likely contribute larger quantities of moisture as the melt extent increases.



## Data availability

Isotopic data is available at https://doi.org/10.18739/A21J9779S. Meteorological data from the Programme for Monitoring of the Greenland Ice Sheet (PROMICE) THU_L site were provided by the Geological Survey of Denmark and Greenland (GEUS) at https://doi.org/10.22008/promice/data/aws. Greenland Ice Sheet melt extent data was provided by Thomas Mote and the NSIDC at https://doi.org/10.5067/MEASURES/CRYOSPHERE/nsidc-0533.001. NCEP Reanalysis sea level pressure data was provided by the NOAA/OAR/ESRL PSL, Boulder, Colorado, USA, at https://psl.noaa.gov/.

## Author contributions

Conceptualization: BGK, PDA, ESK, JMW

Data curation: BGK, PDA, ESK, JMW

Formal Analysis: BGK, PDA, ESK, JMW

Funding acquisition: JMW, ESK

Investigation: BGK, PDA, ESK, JMW

Methodology: BGK, PDA, ESK, JMW

Project administration: ESK, JMW

Resources: BGK, PDA, ESK, JMW

Software: N/A

Supervision: ESK, JMW

Validation: BGK, PDA, ESK, JMW

Visualization: BGK, PDA, ESK, JMW

Writing – original draft: BGK

Writing – review & editing: BGK, PDA, ESK, JMW

## Competing interests

The authors declare that they have no conflict of interest.

## Acknowledgements

This research was supported by the National Science Foundation, Arctic Section, Arctic Observing Program as an EAGER award #1852614 for Arctic water isotope cycle processes and patterns in the Central Arctic during an International Arctic Drift Expedition (MOSAiC) awarded to JMW and ESK. Logistical support and assistance from the United States Air Force, the 821st Air Base Group at Thule Air Base, Vectrus, and Polar Field Services is greatly appreciated. A special thanks for onsite

assistance and maintenance is given to Shawn Arnett, Rich Biggins, Devin Brewer, Joe Burns, David Craig, David Drainer, John Gaston, and Josh Neighbours of the US Air Force, and Michael Pedersen, Henrik Tang Holbek, Michael Kongstedt, and

Gert Mikkelsen of Vectrus. We also thank Picarro for continued technical support for the upkeep of our water vapor analyzer.

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



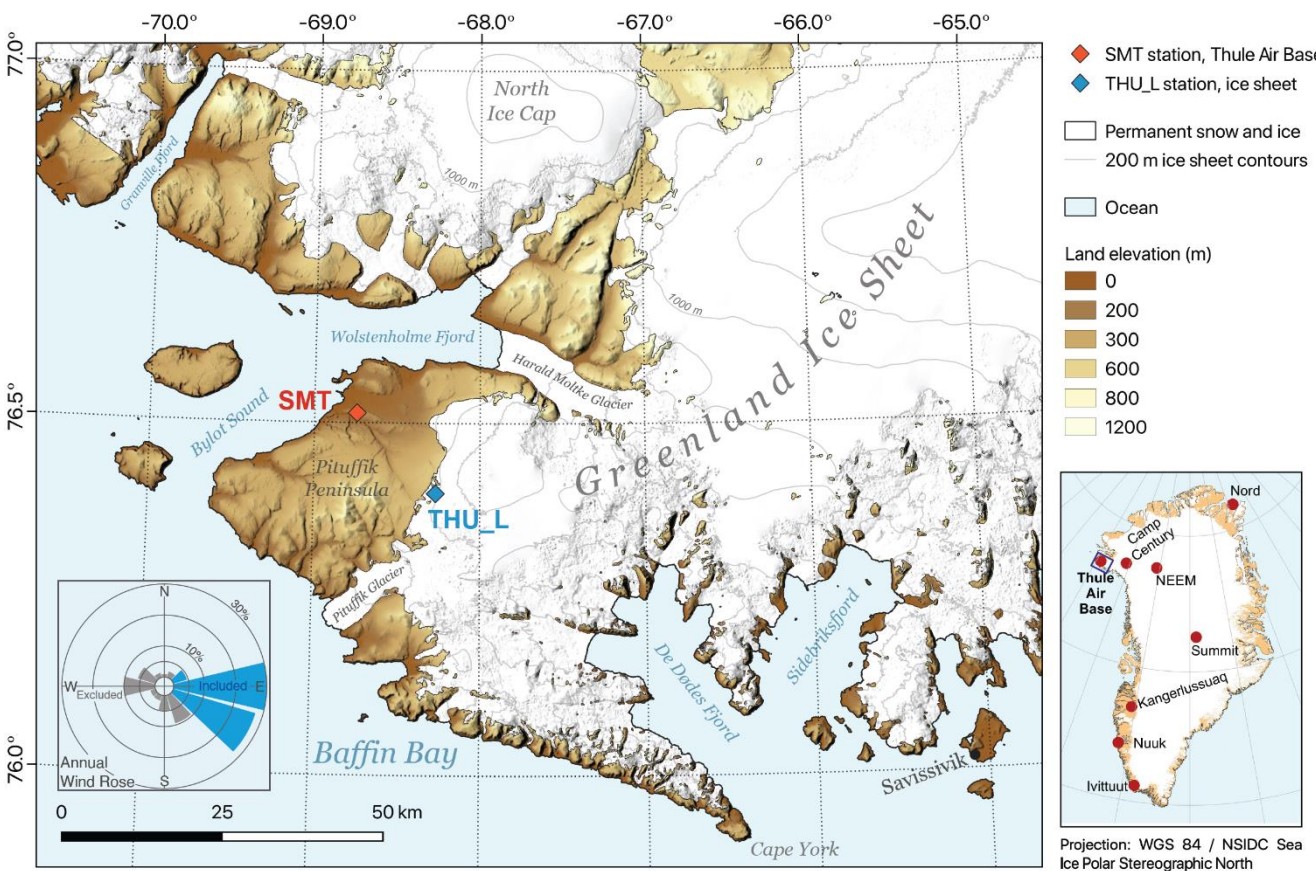

**Figure 1: Map of Thule region displaying location of water vapor isotope measurements at the SMT (red) site in Thule and meteorological measurements to the east over the Greenland Ice Sheet at the PROMICE site THU_L (blue). The insert map shows the location of Thule Air Base and other important locations in Greenland. Elevation data is from ArcticDEM, Polar Geospatial Center (Porter et al., 2018). Ice sheet, land, and ocean extent is from Greenland Ice Sheet Mapping Project, NSIDC (Howat et al., 2014; Howat, 2017). Wind rose created from Thule Airport data covering Aug 2017-Aug 2019 (USAF, 2019).**




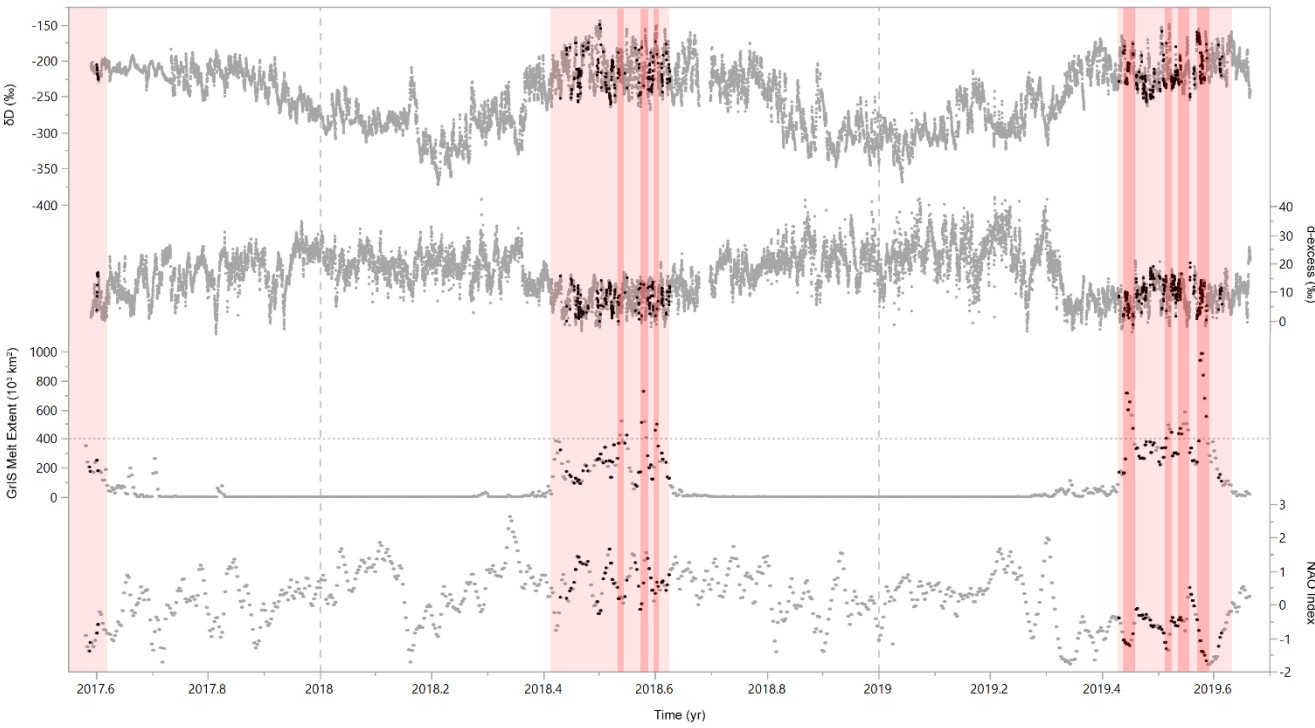

**Figure 2: Time series of a) hourly δD, b) hourly *d-excess*, c) daily GrIS melt extent ($10^3$ km$^2$), and d) daily North Atlantic Oscillation (NAO) Index data. GrIS-vapor as defined by criteria in Sect. 2.3 are shown as black points, with all other data shown as gray points. Time windows defined as melt seasons are highlighted in light red, with major melt events in darker red.**





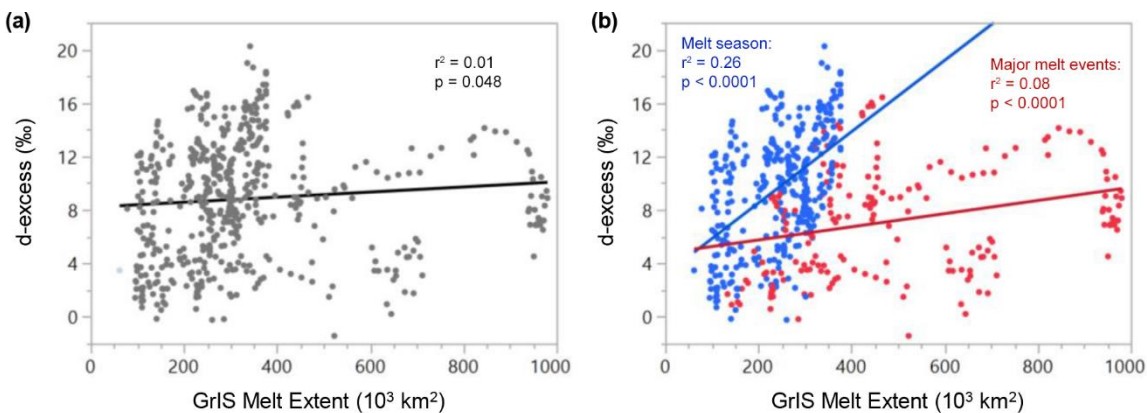


**Figure 3: Linear regression of GrIS melt extent (10³ km²) vs. Thule *d-excess* (‰), where a) the *d-excess* values (gray) include all melt season measurements and b) *d-excess* values are separated by measurements during MMEs (red) and measurements during the rest of the melt season (blue). During the entire melt season (a), the linear regression (black) has an r² = 0.01 and p = 0.048. When separated, the regression for the MMEs (red) has an r² = 0.08 and p < 0.0001, and the rest of the melt season has an r² = 0.26 and p < 0.0001.**








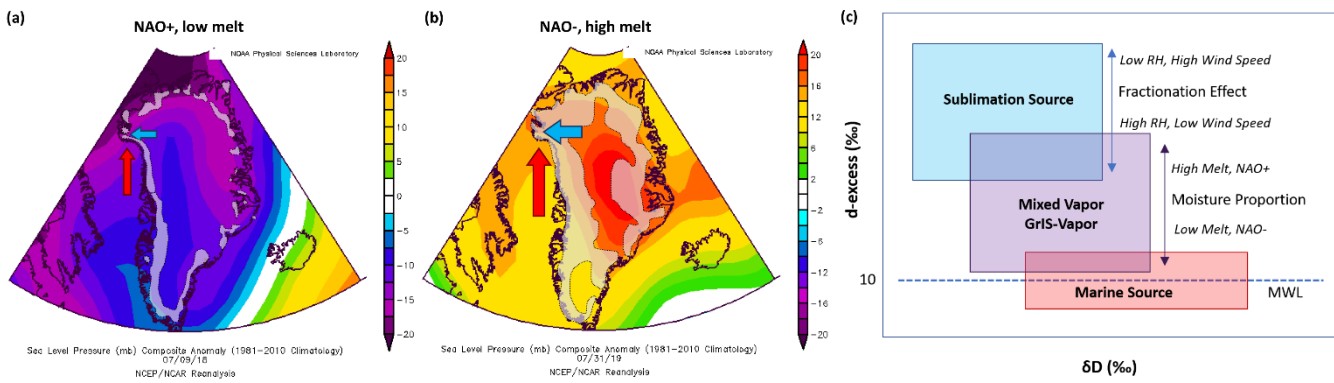

**Figure 4: Theoretical model of primary controls of the water vapor *d-excess* of glacially-derived air (or GrIS-vapor), including mixing model of sublimation- and marine-sourced air that combine to form the measured GrIS-vapor in Thule. (a) NCEP/NCAR Reanalysis (Kalnay et al., 1996) sea level pressure anomaly (mb; displayed as colored contours) during a strong positive NAO on 09 July 2018. Relatively low pressure leads to lower than normal southerly flow up Baffin Bay (red arrow). Relatively low GrIS melt extent (gray) observed on this date, corresponding to low flux of sublimation-sourced vapor (blue arrow). (b) Sea level pressure anomaly (mb) during a strong negative NAO on 31 July 2019. Relatively high pressure leads to higher than normal southerly flow up Baffin Bay (red arrow). Relatively high GrIS melt extent (gray) observed on this date, and thus high sublimation flux (blue arrow). (c) Theoretical mixing model in δD-*d-excess* space of two primary moisture sources, marine (red) and GrIS-derived sublimation (blue), that combine to form the observed water vapor *d-excess* in Thule. The global Meteoric Water Line (MWL) where *d-excess* = 10‰ is shown for reference. The sublimation source *d-excess* is largely determined by the amount of fractionation as water vapor leaves the ice surface, such that low relative humidity (RH) and high wind speed generally yield higher *d-excess* values. The proportion of vapor from the sublimation and marine sources is determined, in part, by the sublimation source area, as approximated by the GrIS melt extent, and the transport of marine moisture, as approximated by the NAO Index. High melt extents and positive NAO phases lead to greater proportion of sublimation-sourced moisture and higher *d-excess*.**



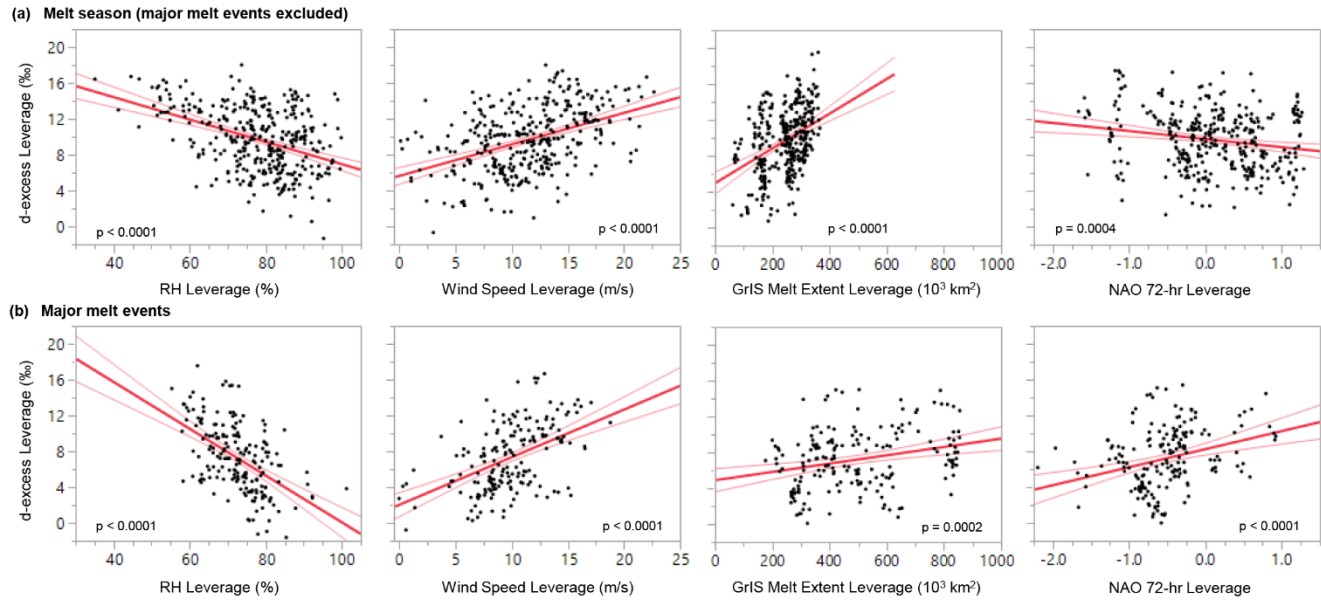

**Figure 5: Leverage plots (Sall, 1990) for multiple regressions of hourly *d-excess* against hourly THU_L relative humidity (RH) and wind speed, past 24-hour GrIS melt extent, the past 72-hour NAO index for (a) the melt season excluding MMEs and (b) only times during MMEs. Thick red line represents best line of fit for each partial regression, and thin red lines are the 0.05 significance curves. The regressions yield overall r$^2$ and p-values of 0.45, <0.0001 and 0.39, <0.0001 for (a) and (b), respectively, and p-values for each individual partial regression are listed on the respective plot.**



**Table 1: Timing of Melt Season and Major Melt Events (MMEs) over length of isotopic measurement period (August 2017-August 2019).**

| Year | Start Melt Season | End Melt Season | Major Melt Events |
|------|-------------------|-----------------|-------------------|
| 2017 | Prior to measurement period | Aug 14 | |
| 2018 | Jun 02 | Aug 17 | Jul 11-21, Jul 30-Aug 03, Aug 07-10 |
| 2019 | Jun 07 | Aug 13 | Jun 11-18, Jul 08-12, Jul 16-23, Jul 28-Aug 03 |