# Peer review of "Significant water vapor fluxes from the Greenland Ice Sheet detected through water vapor isotopic ( $\delta^{18}O$ , $\delta D$ , deuterium excess) measurements"

_The Cryosphere, 2020_

## Referee Comment (RC1) · Anonymous Referee #1 · 10 Dec 2020

General comments: This paper examined the magnitude of Greenland Ice sheet (GrIS) vapor flux based on a 2-year water vapor monitoring data in NW Greenland. Although the data is 2-year long, the authors focused on very short periods to investigate isotopic signature of water vapor derived from the GrIS. The objective of this MS is very interesting. Intuitively, I think it is possible to see the effect of sublimation on the d-excess value. However, my major concern is that there are important issues regarding the multiple regression analysis. In addition, given that the main argument is based on the simple multiple linear regression, the overall length can be significantly shortened. Overall, substantial revision is needed for this MS.

(1) Interpretation of multiple linear regression (Figure 5 and Supplementary figure): The two major variables, humidity and wind-speed are not independent. Those two factors seem to be negatively correlated (i.e., lower humidity under higher wind as generally expected). Thus, the correlation of d-excess vs wind-speed (or vs humidity) may be an apparent correlation without physical mechanism. Generally, multiple regression itself is difficult to apply such situation because of "multicollinearity".

(2) For the regression analysis are based on humidity and windspeed data from THU_L site. Logically, the humidity and wind-speed data in moisture source area should be used. Such data would be obtained by using backward trajectory analysis or other methods. At least, the authors prove that the THU_L data is essentially similar to those of moisture source area.

(3) As described in Section 4.1, the d-excess vs wind-speed relation is complicated. But in general, the relationship is expected to be a non-linear form (Merlivat and Jouzel, 1979).Thus, I don't understand why the author expect a linear relationship between d-excess and wind-speed. In section 4.1, the author noted that "...we anticipate wind speed to have a dominant effect where higher wind speeds cause higher d-excess values whether that is a direct or indirect influence.". I cannot understand the logic of this paragraph. Maybe katabatic wind influence d-excess, but why indirect mechanism results in the linear relationship of d-excess and wind-speed?

Specific comments L167. "A station less than 1 km north" > please specify the distance (i.e., 850 m).

L191-192. "The sublimation source area is likely larger than the melt extent.."> It is often unclear how to separate "sublimation" and "evaporation" for the entire MS. Maybe the d-excess signal you observed is from "evaporation" from melted ice rather than enhanced sublimation.

L.441 "This effect is reduced to 4x greater by including.."> 4 times? The meaning of "reduced to 4 times greater" is not clear.

L.445-448. "It is possible . . ..argues against this" > This long sentence is difficult to understand.

L.449-450. "Meltwater evaporation would likely have a relatively high d-excess compared to that of local marine-source moisture but is lower than sublimated vapor so. . .". >More data is needed to justify this argument.

L465. "It is plausible that the negative phase of NAO. . ."> More data is needed to justify this argument.

L.480."typlically anticorrelated" > Please add citations.

L.525-528. "evaporation has similar effect on the vapor . . ." > A big difference between evaporation and sublimation is that the remaining water is enriched for evaporation but NOT for sublimation. In the case of sublimation, the remaining ice is not mixed, so the isotope ratio is not affected by Rayleigh distillation. Of course, in reality, sublimation in firn layer is much more complicated.
* * *

---

## Referee Comment (RC2) · Anonymous Referee #2 · 26 Jan 2021

Review results for Kopec et al., entitled as Significant water vapor fluxes from Greenland Ice Sheet detected through water vapor isotopic measurement

In this paper, the authors investigated the role of sublimation in the hydrological cycle at Greenland (GL). They have conducted a 2-year monitoring of water vapor isotope and associated meteorological quantities at surface of Thule, northwest of GL, during meting season (around April to June). The authors concluded that the vapor d-excess values are controlled by 1. the surface meteorological condition, i.e., relative humidity and wind speed, and 2. mixing of land and marine derived moisture. According to their end-member analysis, it is told that sublimation contributes 20% of water vapor at the

surface.

First of all, the data they obtained is precious and nice continuous monitoring. It is indeed important to publish their observation data and shared by other researchers. I'd appreciate their effort so much.

However, their analysis is quite primitive. Robust findings are only the correlations between some observed quantities during the target periods. To explain these correlations, they made lots of assumptions, but the explanations are not fully based on recent advancement of the methodology in this field and not quantitative. Therefore, the conclusion is not that convincing. Furthermore, I'm afraid that the conclusion they made (sublimation contributes 20%) is overly exaggerated and such a statement would make the readers misunderstood. By these reasons, I would recommend to reject the manuscript and asking significant revision.

Below, my major comments are listed. I hope that they are somehow helpful to improve this important research.

1. Advanced approach is missing.

For such a topic of hydrological contribution, previous studies have often conducted a budget approach. In case of this study, given surface vapor is influenced by horizontal and vertical moisture fluxes, one can make quantitative analysis to reveal the contribution of these fluxes. Perhaps to do so, some other data like reanalysis product or numerical simulations would be needed. Amount of sublimation can be estimated using typical aerodynamical approach, too. The isotopic observation would play a role to validate such budget analysis. However, the authors did not make any quantitative analysis except the correlations, then they made rather qualitative explanation of uncertainties of the correlations.

2. No fact-based end member analysis

At chapter 5, the authors quantified the contribution of sublimation by the end member

approach. To do this, they used only a mean value of vapor d-excess and referred fixed end-member values. Before then, the authors have made lots of effort explaining the variations of vapor and influencing variable factors, but it was landed on such too rough conclusion. This calculation is not acceptable with at least two reasons. 1. there is no evidence that the fixed values of two end members are suitable for this case. 2. the contribution should be different in each time (thus the d-excess of vapor varies a lot) and the mean of contribution cannot be calculated from the mean values of observations. Contrast to this premature estimation, the number obtained here (20%!) has strongly impactful message.

3. Possibility of marine-derived vapor

According to previous studies, clear negative relationship between RH and vapor d-excess can be seen over oceans. Probably surrounding of GL, too. If so, there is possibility that both the RH and d-excess vapor at Thule are influenced by the vapor evaporated from the surrounding oceans, not from sublimated vapor. In this regard, mixing of land and marine derived vapor is indeed important as the authors stated. Before analyzing with NAO, more straightforward approach, such as water budget, should be used to quantify the mixing.

4. Isotopic evidence of evaporation and sublimation

Is it indeed true that water isotope observation can distinguish sublimation (solid to vapor) and evaporation (liquid to vapor)? Of course the fractionation factors for these two processes are different, but the difference caused by these really detectable from this observation? Furthermore, I don't know the reason why the contribution of sublimation should be separated from evaporation. Both use latent heat. The target period of this study is melting period. Evaporation from the melted water is essentially the same as sublimation in energy budget point of view. If the ice mass loss is the target, there is no need to distinguish sublimation and evaporation, I guess.